# The 2013–2018 Matese and Beneventano Seismic Sequences (Central–Southern Apennines): New Constraints on the Hypocentral Depth Determination

**Brando Trionfera [1], Alberto Frepoli [2,\*], Gaetano De Luca [2], Pasquale De Gori [2] and Carlo Doglioni [1,2]** 

[1]  Dipartimento Scienze della Terra, Università La Sapienza, 00185 Roma, Italy; brando.trionfera@gmail.com (B.T.); carlo.doglioni@ingv.it (C.D.)

[2]  Istituto Nazionale di Geofisica e Vulcanologia, Osservatorio Nazionale Terremoti, 00185 Roma, Italy; gaetano.deluca@ingv.it (G.D.L.); pasquale.degori@ingv.it (P.D.G.)

\*  Correspondence: alberto.frepoli@ingv.it

**Abstract:** The Matese and Beneventano areas coincide with the transition from the central to the southern Apennines and are characterized by both SW- and NE-dipping normal faulting seismogenic structures, responsible for the large historical earthquakes. We studied the Matese and Beneventano seismicity by means of high-precision locations of earthquakes spanning from 29 December 2013 to 4 September 2018. Events were located by using all of the available data from temporary and permanent stations in the area and a 1D computed velocity model, inverting the dataset with the Velest code. For events M > 2.8 we used *P*- and *S*-waves arrival times of the strong motion stations located in the study area. A constant value of 1.83 for $V_p/V_s$ was computed with a modified Wadati method. The dataset consists of 2378 earthquakes, 18,715 *P*- and 12,295 *S*-wave arrival times. We computed 55 new fault plane solutions. The mechanisms show predominantly normal fault movements, with T-axis trends oriented NE–SW. Only relatively small E–W trending clusters in the eastern peripheral zones of the Apenninic belt show right-lateral strike-slip kinematics similar to that observed in the Potenza (1990–1991) and Molise (2002 and 2018) sequences. These belong to transfer zones associated with differential slab retreat of the Adriatic plate subduction beneath the Apennines. The Matese sequence (December 2013–February 2014; main shock $M_w$ 5.0) is the most relevant part of our dataset. Hypocentral depths along the axis of the Apenninic belt are in agreement with previous seismological studies that place most of the earthquakes in the brittle upper crust. We confirm a general deepening of seismicity moving from west to the east along the Apennines. Seismicity depth is controlled by heat-flow, which is lower in the eastern side, thus causing a deeper brittle–ductile transition.

**Keywords:** seismogenic structures; hypocentral determination; 1D velocity model; central–southern Italy

## 1. Introduction

An accurate hypocentral determination is an important pre-requisite in seismological investigation and geodynamic interpretation. In the last 15 years, the high number of temporary and permanent seismic stations in the Italian region represents the progress made toward this goal. Considering the established seismotectonic pattern of the southern Apennines extensional belt, we present evidence of three seismic sequences in the Sannio–Matese area (central–southern Italy), with main shock and aftershock hypocentral depths in agreement with previous seismological studies. The large 1980 Irpinia earthquake ($M_w$ 6.9), caused by a slip on a NE-dipping normal fault, provides valuable parametric information and inference on the geometry and depth of the seismogenic faults in the southern

Apennines [1–3]. This earthquake released most of its energy between a depth of 3 km and 12 km [4]. Similarly, the NE-dipping Boiano fault responsible for the 1805 $M_w$ 6.6 event [5–7] shows the same geometry at a depth suggesting an energy release at the same depth range. Other faults associated with individual historical earthquakes in the southern Apennines are thought to extend down to 15–16 km of depth, which is commonly the maximum nucleation depth as shown by the moderate to large seismicity in the last 35 years. Moreover, this observation is consistent with the 15–18 km depth of the brittle–ductile transition (BDT) in the crust beneath the central–southern Apennines [8,9]. Within the southern Apennines extensional belt, particularly in the internal section, there are also SW-dipping faults, possibly associated with some destructive historical and prehistorical events. No low-to-moderate events occurred on these faults in the instrumental era, until the recent Matese sequence (2013–2014). Among the historical seismicity, only the $M_w$ 6.6 September 1349 earthquake (*Aquae Iuliae* fault, northwest Matese massif), investigated with paleoseismological and archeoseismological data by Galli and Naso (2009) [10], shows an SW-dipping extensional seismogenic structure.

In this study, we show that the $M_w$ 5.0 29 December 2013 earthquake was nucleated along a NW-striking and SW-dipping normal fault at 12.3 km depth, that borders the southwestern part of the Matese massif. In the northeastern part of the massif, during 2016–2017, two sequences occurred along NE-dipping normal faults, consistent with the geometry of previous historical and instrumental seismicity. We extended the seismicity study to the Beneventano area and the eastern Molise region collecting data until September 2018. We investigated the seismicity in the studied area with all the seismic data available, confirming the quality and stability of the locations using a valid 1D velocity model and an appropriate $V_p/V_s$ ratio for this region.

*Geologic and Seismotectonic Setting of the MATESE and Beneventano Areas*

The Matese and Beneventano areas are located within the southern Apennines seismogenic belt, around 40 km wide, that is nearly continuous along central–southern Italy for several hundreds of kilometers. This belt is composed of a number of 30–40 km-long neighboring fault segments responsible for the large magnitude ($M_w \leq 7$) historical earthquakes. Along these faults, almost pure dip–slip movement accommodates the NE–SW extension perpendicular to the axis of the Apenninic belt, with a rate of about 3–4 mm/yr [11–14] (Figure 1). This NE-trending extension began in the late Miocene [15–17] and is related to the subduction hinge migration away from and relative to the upper plate [18,19]. As a result, the Tyrrhenian back-arc basin opened and the upper mantle is kinematically required to compensate the slab retreat at depth [20,21]. As a result of previous compression activity, the active NW–SE striking extensional faults are superposed into earlier thrust and transpressional deformation [22–24]. Being shallower the mantle in the Tyrrhenian side, the western Apennines have a heat flow of around 60–80 mW/m$^2$, whereas it decreases to 30–40 mW/m$^2$ in the central and eastern side of the Apennines and in the study area [25] above the subduction hinge. Therefore, we expect a brittle thickness of the upper crust of about 8–10 km in the western side of the Apennines, and an increase to 16–18 km in the eastern side and in the study area [8]. Moreover, due to the subduction, a deepening of the brittle–ductile transition is expected beneath the belt (Figure 1).

The Mesozoic–Cenozoic carbonate platform and active margin-related siliciclastic rocks formed the Matese massif [26]. This mountain range is bound to the southwest by the alluvial Volturno Basin and to the northeast by the Boiano Plain, a 30 km-long NW–SE elongated basin, mainly filled with Neogene terrigenous deposits [27]. The Matese carbonate platform rocks are thrust above the coeval basin carbonate and terrigenous rocks of the Sannio–Molisan thrust assemblage, which is extensively exposed to the east [28]. In turn, the Sannio–Molisan basin rocks rest structurally above the Mesozoic–Cenozoic carbonate platform rocks, which are the westward continuation of the Apulia platform exposed further to the east in the foreland of the Apennines.

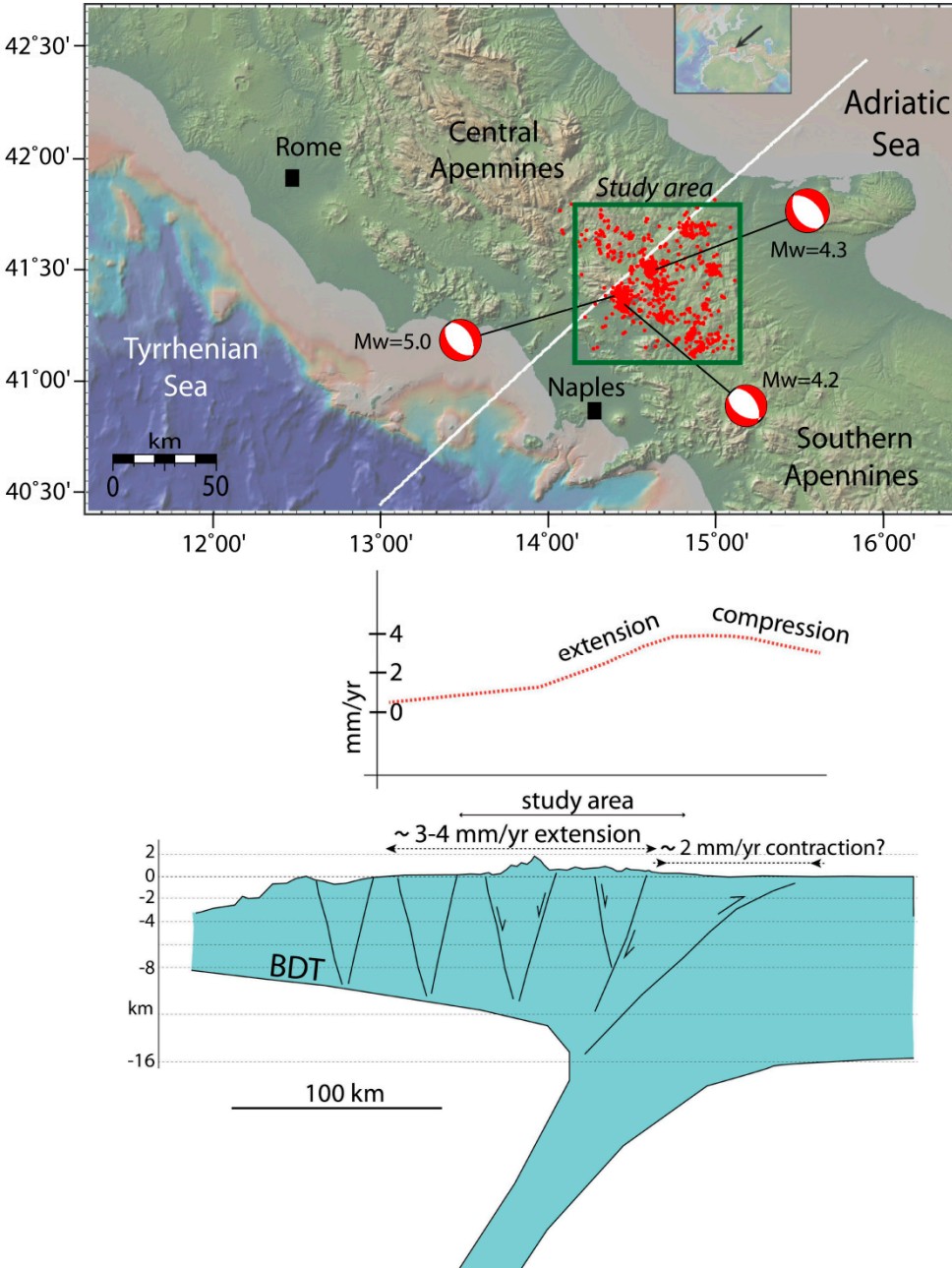

**Figure 1.** Area of the central–southern Apennines (green outline) and studied seismicity (red dots) with fault plane solutions of the most relevant events of this study. Lower part, velocity profile (NE–SW) across the central–southern Apennines from GPS data [14].

The Matese and Beneventano areas are characterized by a high seismic hazard (Figure 2). Several seismotectonic studies have been performed for these areas. Some of these studies have made use of methodologies for the recognition of structural features in active tectonic areas integrating morphological and morphometrical data derived by DEMs (digital elevation models) with different geophysical data [29,30]. Moreover, an integrated analysis of structural, seismic and gravimetric data for the identification of active fault geometries was performed by Gaudiosi et al., in 2015 [31]. Based on historical earthquakes records, the areas have suffered intensity X or higher over past centuries [32]. In the last 35 years, seismicity has been recorded as low-to-moderate activity, with just a few events having a magnitude larger than $M_w$ 5.0 [33].

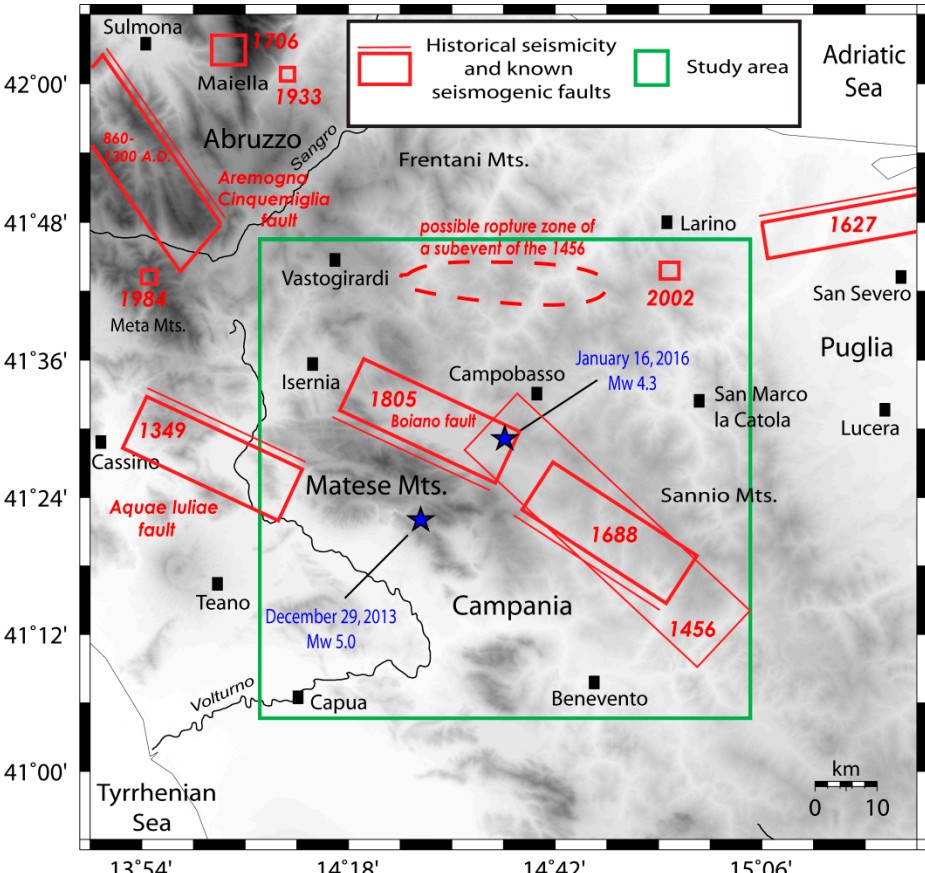

**Figure 2.** Map showing the investigated area in the central–southern Apennines (green outline). Historical earthquakes after CPTI15 [32] and seismogenic structures (modified from Valensise et al., 2004 [34]). The presumed rupture zone of a possible subevent of the 1456 sequence is outlined with a red dashed ellipse.

From a tectonic point of view, the Matese area coincides with the transition zone between the central and southern Apennines fold-and-thrust belt. This is a transfer zone characterized by divergent extensional seismogenic fault systems [35]: faults in the central Apennines dip to the southwest; those in the southern Apennines dip to the northeast.

In particular, the Matese area is dissected by a NW–SE striking dip–slip normal fault and is characterized by both NE- and SW-dipping seismogenic structures. The SW-dipping structures uplift the range relative to the Volturno Basin including the 9 September 1349 $M_w$ 6.6 event [10]. To the northeast, the strong $M_w$ 6.6 1805 earthquake [5,36] shows a NE-dipping fault. For this historical event, no surface breaking was reported. A detailed geological–structural analysis of the northern slope of the Matese massif defines the geometry and segmentation patterns of the NE- to N-dipping fault sub-systems that belong to the North Matese Active Fault System [37]. Southeast of Matese, in the Beneventano area, an $M_w$ 7.0 earthquake occurred in 1688 [38,39]. The isoseismal reconstructed by Serva (1985) [38] shows a strong elongation parallel to the axis of the affected sector of the Apenninic chain [40]. Nappi et al., (2008) [29] studied the possible location of the 1688 seismogenic source. They have selected a lineament which strike shows coherence with the 1688 fault geometry [41]. To the east and the northeast of the Matese massif, within the Adria plate, E–W strike–slip faults of moderate-to-low seismicity were observed [42,43]. Many authors consider these structures as potential seismogenic sources of moderate-to-large magnitude events [34,44] as shown in Figure 2 (red dashed ellipse) for the presumed subevent of the 1456 sequence.

Finally, in the internal part of the massif, the SW-dipping Matese Lake fault is suspected to have slipped during the Holocene [45,46], but there is not clear geomorphic evidence of this activity along

this tectonic feature. The extensional structures of the southern Apennines are known mainly at the surface or in their shallow expressions. Brozzetti (2011) [24] investigated the deep geometry of these structures by integrating new field data with seismic lines interpretation.

## 2. Methods

### Data Collection and Analysis

The aim of this paper is the recovery of all the seismic data available from permanent (Seismic National Network, RSN) and temporary stations of the Istituto Nazionale di Geofisica e Vulcanologia (INGV), the regional network of Abruzzo (RSA; [9,47] and the national accelerometric network (RAN) in the central–southern Apenninic region, in order to get the best locations of the seismicity in the study area (Figure 3). All the waveforms of the Matese–Beneventano seismic sequences were re-picked to obtain both more accurate arrival times of *P*- and *S*-waves and a larger number of first-motion polarities, as compared to the routine procedure carried out in the compilation of seismic bulletins.

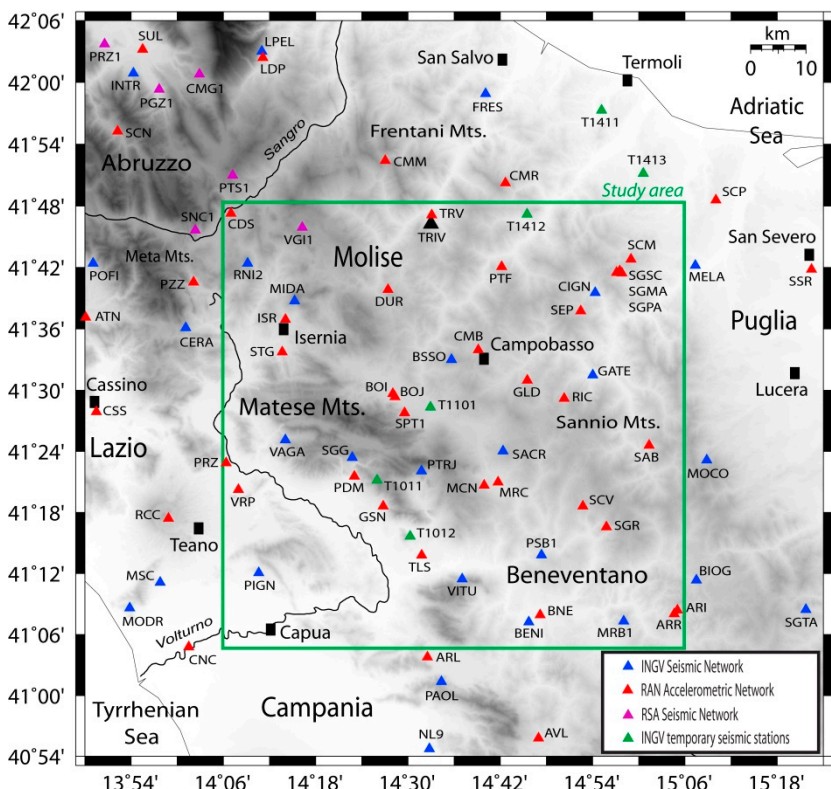

**Figure 3.** Seismic networks used in this study (see legend on the lower right). Temporary stations T1011 and T1012 (in dark green) wereinstalled after the mainshock $M_w$ 5.0 (29 December 2013), and T1101 after the mainshock $M_w$ 4.3 (16 January 2016).

On 30 December 2013, one day after the $M_w$ 5.0 San Gregorio Matese main shock, the Rete Mobile of INGV installed two temporary stations in the epicentral area of the sequence, that operated until 4 March 2014 [48]. Similarly, after the $M_w$ 4.3 main shock of Baranello, on 17 January 2016, INGV installed one temporary station in this locality for two months [49]. The waveforms of all the earthquakes, used to pick arrival times and to determine the hypocenter, were extracted from the continuous data stream of the three temporary stations and merged with those available from the permanent network RSN. This collection was performed by using the trigger time based on standard STA/LTA ratio methods provided by the RSN. In addition, during the early days after the $M_w$ 5.0 main shock and the $M_w$ 4.2 aftershock of the first sequence, the continuous recording of the permanent station VAGA (Valle Agricola, Figure 3) was visually scanned to recover all the aftershocks. Moreover,

for the strongest events (M > 2.8), we used the waveforms of the accelerometric network (RAN) installed in the epicentral area and the stations of the Regional Seismic Network of Abruzzo (RSA) located to the north and northwest of the Matese Range. With manual picking, we assigned a weight to each time pick according to the following scheme [9]: weight 100% for reading errors less than 0.04s, 75% for errors between 0.04 and 0.1s, 50% for errors between 0.1 and 0.2s, and 25% for errors between 0.4 and 1.0s. Picking with errors higher than 1.0s were not used in the relocation.

We detected 2378 earthquakes from 29 December 2013 to 4 September 2018; 362 of these earthquakes were rejected due to their low location quality (hypocentral errors larger than 2 km; Hypoellipse code, [50]). The whole dataset consists of 18,715 *P*- and 12,295 *S*-wave arrivals. Compared to the number of events located only by the RSN during the same period (1474), we detected 904 more events, approximately 38% more. The average RMS calculated on the residuals of the 2016 selected events is 0.08 s.

To compute the best 1D velocity model, we selected 1287 events that satisfied the following selection criteria: at least 6 *P*- and 2 *S*-phases, hypocentral errors less than 2 km, azimuthal gap less than 180° and the nearest station within 10 km from epicenter. We applied an area selection in order to get a homogeneous distribution of the ray sampling. We divided the crustal volume in the depth range 0–15 km in 3 km cube cells and we retain only one event for each sub-volume. After that, we collected 798 events covering the whole area. Seismicity occurring at greater depth (z>15 km) did not require spatial selection, since the number of events was significantly lower (489).

We inverted the selected dataset with the Velest code [51] starting from 7 different velocity models. These velocity models have been chosen among published works with relocated central–south Apennines earthquakes [9,52–57]. Earthquakes of our dataset were relocated with the new output 1D *MATESE* velocity model (Figure 4a; Table 1) and the Hypoellipse code [50]. A decreasing weight with distance was used (weight 1 in the 0–30 km range, decreasing to 0 from 30 km to 60 km). We used a constant value of 1.83 for $V_p/V_s$ determined with the Wadati method [58] (Figure 4b). $V_p/V_s$ was computed using all the available events of the database, coupling the best *P*- and *S*-arrival times (weight 0–2).

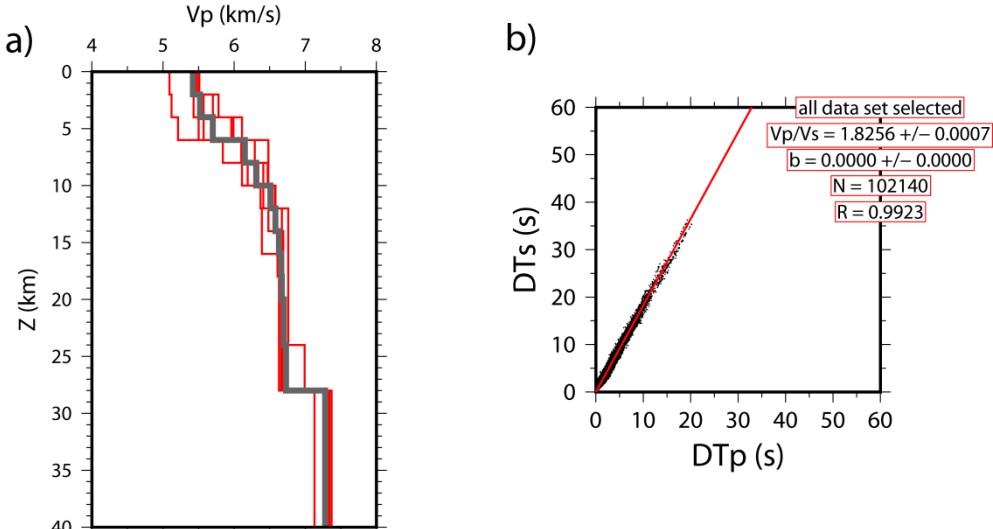

**Figure 4.** (**a**) The 1D *MATESE* velocity model computed with the Velest code [51]; (**b**) Vp/Vs Wadati diagram computed using all the available events of the database coupling the best *P*- and *S*-arrival times (weight 0–2).

**Table 1.** Final 1D *MATESE* model computed through the inversion of the selected dataset with Velest code [51].

| 1D MATESE | |
| --- | --- |
| Vp | depth |
| 5.42 | 0.00 |
| 5.52 | 2.00 |
| 5.70 | 4.00 |
| 6.15 | 6.00 |
| 6.31 | 8.00 |
| 6.51 | 10.00 |
| 6.58 | 12.00 |
| 6.63 | 14.00 |
| 6.66 | 16.00 |
| 6.67 | 18.00 |
| 6.70 | 20.00 |
| 6.73 | 24.00 |
| 7.28 | 28.00 |
| 8.10 | 40.00 |

The upper part of the 1D *MATESE* velocity model (0–6 km) is characterized by the alternation between carbonate rocks and basin sequences. $V_p$ values are ranging between 5.4–5.7 km/s. At intermediate depths (6–12 km) a more homogeneous trend is evident. This part coincides with the high-strength zone at seismogenic depth beneath the Matese limestone platform identified by a tomographic study [59]. $V_p$ values range between 6.3–6.6 km/s. The 5–10 km depth interval coincides with the Meso–Cenozoic succession of the Apulia and western Carbonate Platform according to the geological model proposed by Mostardini and Merlini (1986) [60]. A further increase in speed (6.6 km/s; 12 km of depth) is evident in the lower part of the Apulia carbonate multilayer which corresponds to the Triassic dolomites and the Anhydrites of the Burano formation. The sedimentary cover thickness gradually decreases between 9 km and 11 km of depth [61]. In the deepest part of the model (14–28 km), within the crystalline basement, we have a further increase in speed (from 6.6 to 7.3 km/s).

As a result of the 1D *MATESE* velocity model and to the high station coverage, we were able to determine all earthquake hypocenter depths with acceptable uncertainties. Within the Matese and Beneventano area, the average station spacing is around 8 km. The average location errors for the events are 0.18 km horizontally and 0.73 km vertically, with a confidence level of 90%.

For our selected dataset hypocenter depth extends down to 32 km with a pronounced maximum between 6 km and 14 km, except for a deep $M_L$ 3.7 earthquake at 335 km depth, detected in the Macchiagodena sector (CB), being evidence of the active subduction of the Adriatic plate beneath the southern Apennines.

## 3. Results

All the study seismicity belongs to low-to-moderate magnitude sequences and swarms. In the following section, we describe the main characteristics of the seismicity pattern and the fault plane solutions computed, starting from the Matese 2013–2014 sequence, which represents the most substantial and important part of this work. In all the maps and cross-sections, we show only the relocated seismicity with output quality A of Hypoellipse (hypocentral errors less than 1 km; 2016 selected events; Figure 5).

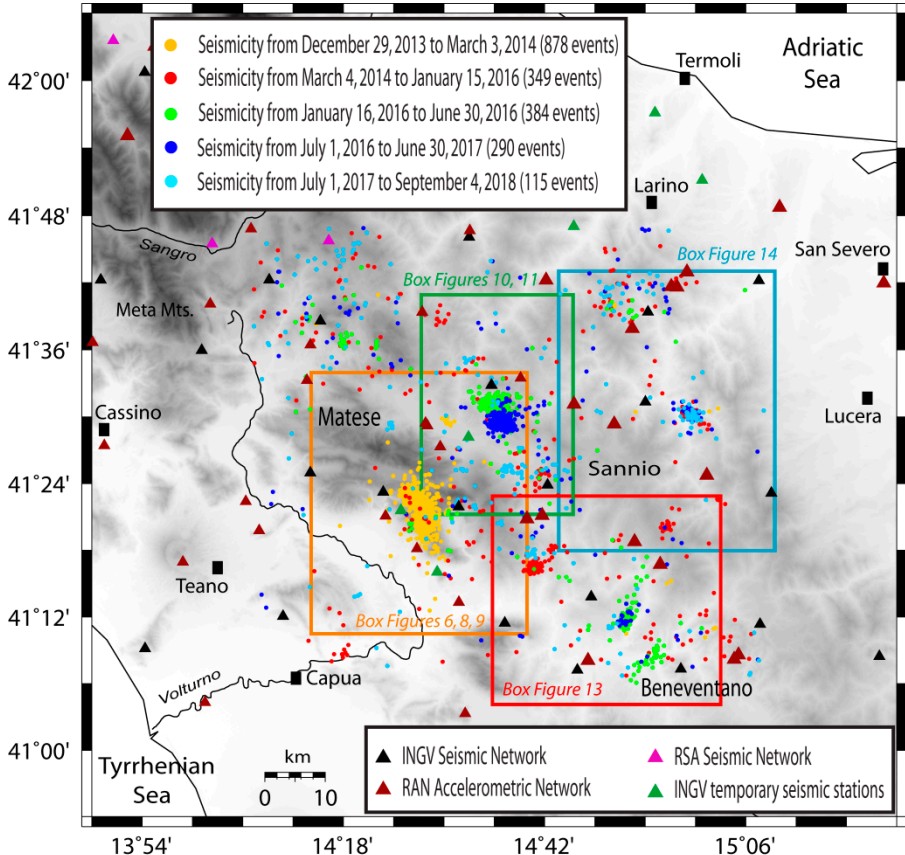

**Figure 5.** Map showing the epicentral locations of the 2016 earthquakes selected for the analysis. Events are separated into five periods (see legend in the upper part). Colored boxes show the areas included in the next figures with their seismicity.

### 3.1. The Matese Sequence

On 29 December 2013, a magnitude $M_w$ 5.0 earthquake occurred close to the localities of San Gregorio Matese and Piedimonte Matese (Caserta province). The event was generated along a NW-striking and SW-dipping normal fault that borders the southern part of the Matese Range [48]. It was preceded by a few foreshocks, the largest one with a magnitude of $M_L$ 2.7. Centroid-moment tensor solutions [62] (http://www.eas.slu.edu/eqc/eqc_mt/MECH.IT/, [63]) and the distribution of aftershocks define a fault plane that dips around 50° between depths of 10 kmand 15 km. This sequence was characterized by the largest aftershock, which occurred on20 January 2014 $M_w$ 4.2, and activated a southern cluster close to the locality of San Potito Sannitico. The sequence lasted for two months and was characterized by a substantial decay during February 2014.

We detected 926 aftershocks and 878 are the relocated events with quality A (Hypoellipse output; hypocentral errors less than 1 km; Figure 6). According to the temporal distribution of earthquakes, the sequence is separated into two parts (Figure 7), both located in the southern portion of the Matese massif. The first one belongs to the $M_w$ 5.0 mainshock, while the second one to the largest aftershock $M_w$ 4.2. These two events have hypocenters at 12.3 km and 12.7 km respectively (hypocentral error, ±0.1). Four low magnitude foreshocks occurred a few hours before the mainshock. Aftershocks show a hypocentral depth between 4 km and 16 km, but mostly concentrated in the 8–15 km depth range. Nine aftershocks have a magnitude $M_L$ ranging from 3.0 to 4.2. For these events we also have the RAN stations data. A total of 65 aftershocks have a magnitude $M_L$ from 2.0 to 2.9, while a large number of events lie within the magnitude range of $M_L$ 0.4–1.9.

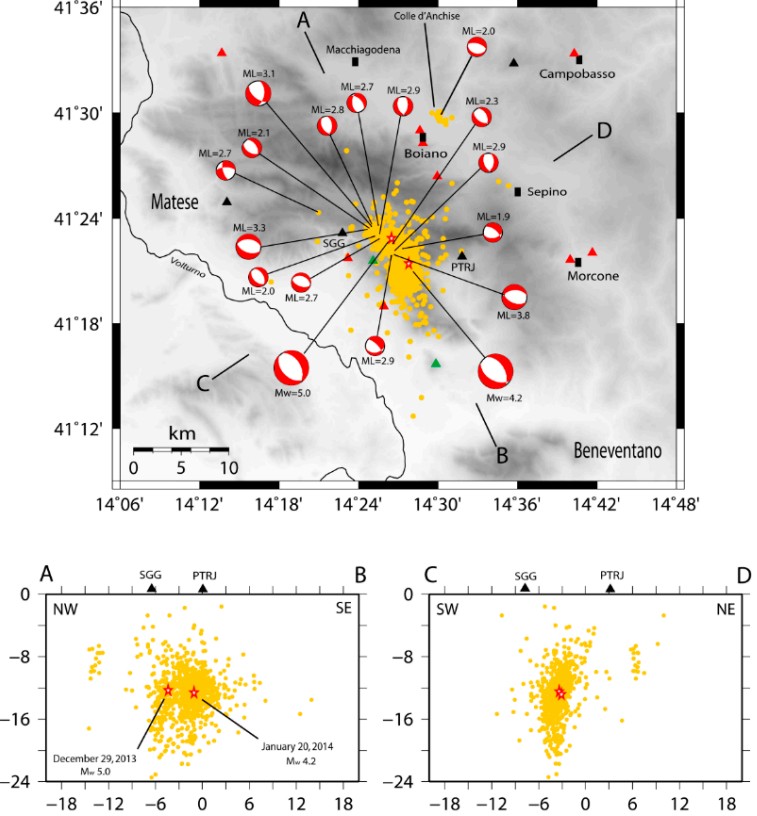

**Figure 6.** Hypocentral locations with the Hypoellipse code and fault plane solutions of the Matese 2013–2014 seismic sequence. Red stars indicate hypocenters of the mainshock ($M_w$ 5.0) and the largest aftershock ($M_w$ 4.2). Cross-section width is 30 km, as in all the next figures.

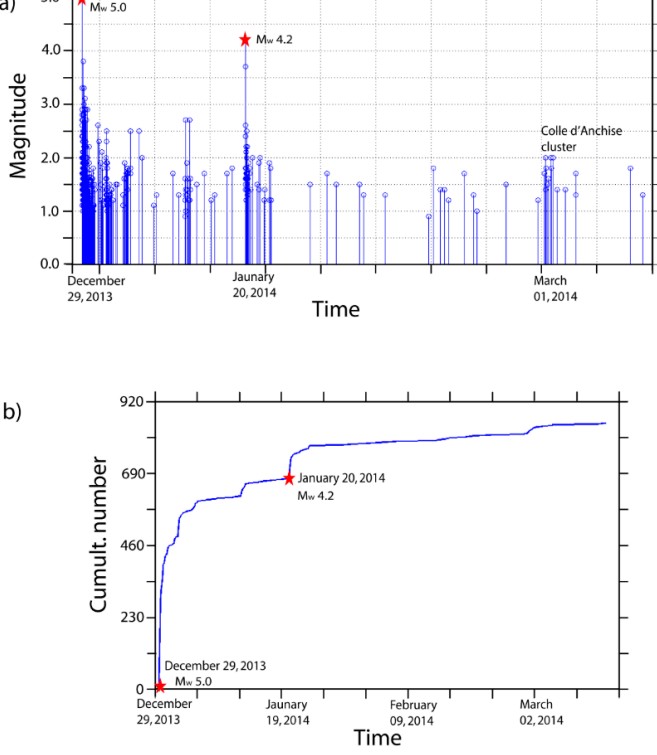

**Figure 7.** Matese 2013–2014 seismic sequence: (**a**) magnitude vs. time diagram; (**b**) event cumulative number diagram.

The cluster of the whole sequence is stretched in an NNW–SSE direction and it shows the largest concentration of aftershocks in its southern part (Figure 6). In fact, in the cross-section AB (along-strike) of Figure 6, we observe a dense cloud of earthquakes exactly around the hypocenter of the largest aftershock $M_w$ 4.2, with a depth ranging from 8 km to 17 km. In the first part of the sequence, aftershocks are concentrated mostly in the NW part of the cluster at adepth between 10 km and 16 km. This feature is clearer when observing the map and cross-sections of plot a in Figure 8. It shows the seismicity in the first two days (29–30 December 2013) in which seismicity is concentrated into two clusters, both with a depth ranging between 10 km and 16 km. These two clusters delineate a small seismic gap (cross-section AB of plot a in Figure 8). This gap is filled by aftershocks in the following days, as shown in cross-section AB of Figure 6.

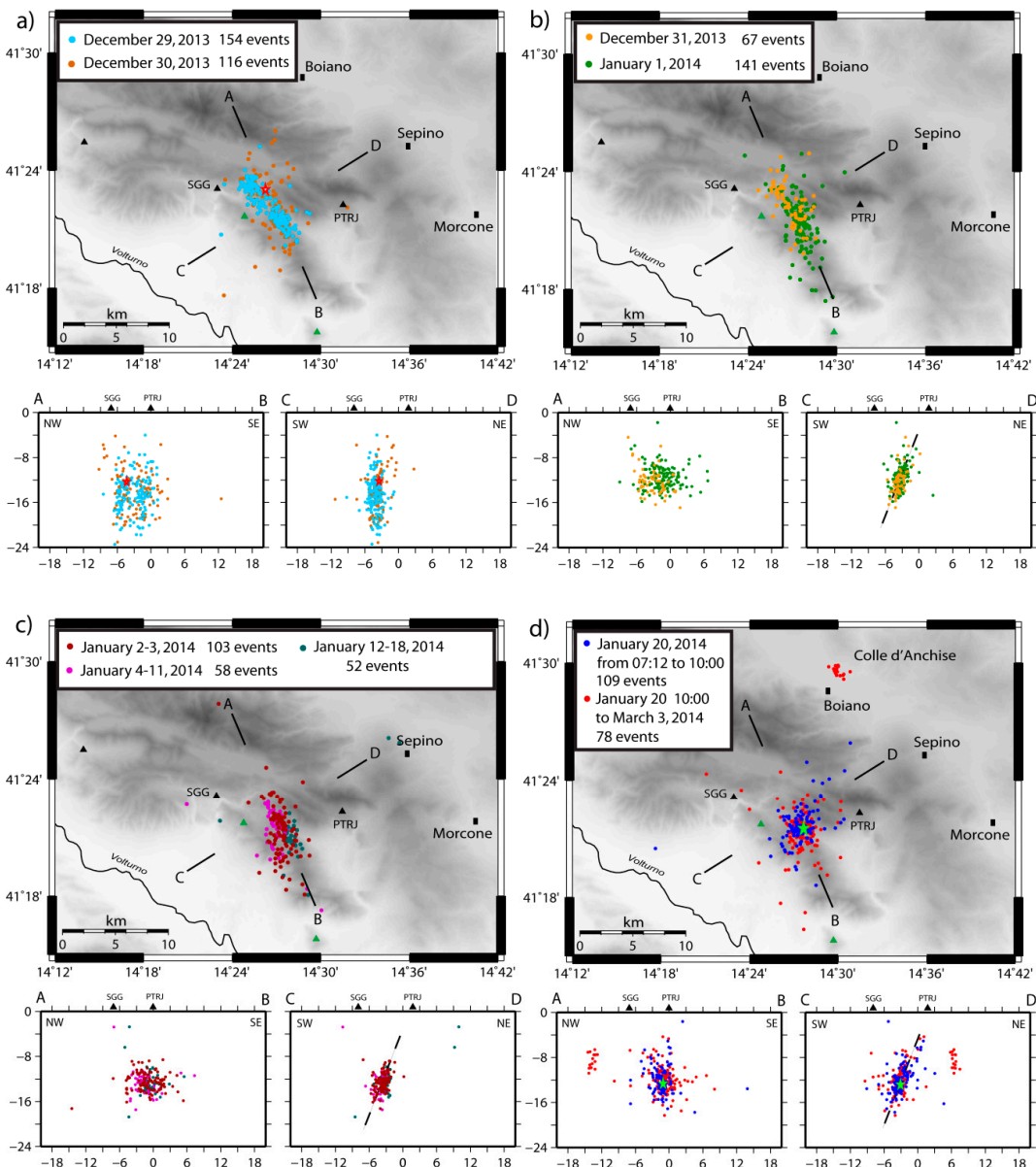

**Figure 8.** Space–time evolution of the Matese 2013–2014 seismic sequence (878 events) relocated with the Hypoellipse code; (**a**) seismicity of 29–30 December 2013; (**b**) seismicity of 31 December 2013–1 January 2014; (**c**) seismicity of 2–11 January 2014; (**d**) seismicity of 20 January–3 March 2014. Red star in plot (**a**) indicates the main shock $M_w$ 5.0, while green star in plot (**d**) indicates the largest aftershock $M_w$ 4.2.

In the two first days of this sequence, before the installation of the two temporary stations in the epicentral area (the first one in the early afternoon of the 30 December), we observe many low magnitude aftershocks located in the 16–22 km depth range, with small hypocentral errors (quality A of Hypoellipse) but with the nearest station (VAGA) only at 15–16 km from the epicenter, and for this reason, the locations are rejected. In fact, a small formal Hypoellipse error does not necessarily imply a good hypocentral location. Event locations obtained with a small number of stations not very close to the epicenterusually give a poor estimate of the real depth. In these cases, it is necessary to use additional selection criteria, i.e., the nearest station selection criteria.After the installation of the two temporary stations, there is a general improvement in the location quality of the seismicity and noadditionaldeeper events are located, as shown in plot b, c and d of Figure 8.

Seismicity moves to the S–SE during the first 20 days of January 2014. Plots **b** and **c** of Figure 8 show a concentration of seismicity in the area hit by the 20 January 2014 $M_w$ 4.2 earthquake. From 14 Februaryto 1 March 2014, a small and low magnitude ($M_L$ maximum 1.8) swarm is located around 12 km to the N–NE of the Matese sequence (plot d in Figure 8), near the Colle d'Anchise locality (CB), an area hit on 16 January 2016 by the $M_w$ 4.3 main shock of the Baranello sequence. This small cluster delineates a plane dipping to the NE, with depth between 4 km and 12 km (cross-section CD in plot d of Figure 8).

We performed double-difference locations [64] considering only the aftershocks located using the temporary stations (closest stations to the epicentral area) and very close in time to the main shock and the largest aftershock of this sequence (Figure 9). A good result with the HypoDD code is achieved only with a homogeneous group of events, i.e., events that are very close to each other and localized with the same group of stations. The first group of 106 events belongs to the second half of 30 December and to the whole day of 31 December 2013 (Figure 9a). The second group consists of 196 events of 20 January 2014, the day of the largest aftershock ($M_w$ 4.2; Figure 9b). Cross-sections CD in both figures show clusters dipping to the SW with a high angle (around 70°). The largest aftershock's cluster (20 January) is slightly located to the S–SE.

We found 88 best fit double-couple focal mechanisms for the whole study area using the fault plane fit (FPFIT) grid-search algorithm of Reasenberg and Oppenheimer (1985) [65]. *P*-wave polarities were determined manually and picked on the raw seismograms. Out of these 88 computed solutions, we selected 55 focal mechanisms well constrained by 12 or more observations homogeneously distributed on the focal sphere. To select the most constrained solutions, we followed the FPFIT values of $Q_f$ and $Q_p$ output quality factors (Table 2). $Q_f$ gives information about the solution misfit of the polarity data $F_j$, whereas $Q_p$ reflects the solution uniqueness in terms of the 90% confidence region for the strike ($\Delta$s), dip ($\Delta$d) and rake ($\Delta$r). The quality factors range from A to C for decreasing quality. All focal mechanisms with one or both quality factors C, and with <12 observations, were rejected. The majority of the selected focal mechanisms represent pure normal faults and normal faults with a small strike–slip component. Only the two clusters of Foiano in Val Fortore and San Marco la Catola, located in the eastern sector of the study area, show almost pure strike–slip focal mechanisms. Figure 6 shows the 16 fault plane solutions computed for the Matese 2013–2014 sequence. The orientation of the T-axis of these normal solutions clearly shows that the Matese massif is affected by an extensional stress regime along NE–SW.

**Table 2.** Quality factors for fault-plane solutions (fault plane fit (FPFIT); [65]).

| Quality | $Q_f$ | $Q_p$ |
|---------|-------|-------|
| A | $F_j \leq 0.025$ | $\Delta$s, $\Delta$d, $\Delta$r $\leq 20°$ |
| B | $0.025 < F_j \leq 0.1$ | 20–40° |
| C | $F_j > 0.1$ | >40° |

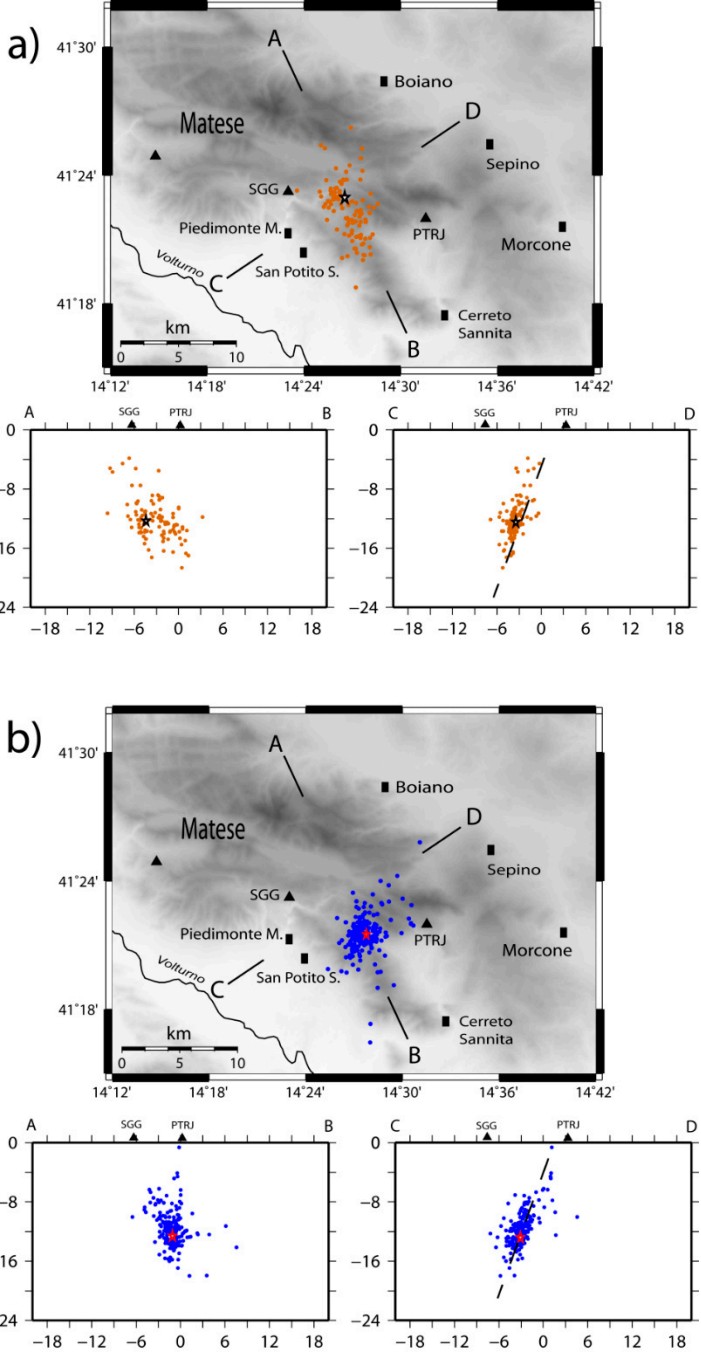

**Figure 9.** (**a**) Double-difference locations (HypoDD) of the 30–31 December 2013 aftershocks. Only events located with the two temporary stations are used (106 events). Black star shows the $M_w$ 5.0 main shock location. (**b**) Double-difference locations (HypoDD) of the 20 January 2014 aftershocks (196 events). Red star shows the largest aftershock ($M_w$ 4.2) location.

### 3.2. The Baranello Sequence and the Vinchiaturo Swarm

On 16 January 2016, the northeastern Matese area was hit by an $M_w$ 4.3 earthquake close to the locality of Baranello at 10 km hypocentral depth [49]. This event was located around 16 km to the NE of the Matese 2013–2014 sequence. This main shock was preceded by 34 foreshocks from 10 to 16 January, with a maximum magnitude of $M_L$ 2.9. The foreshocks delineate a small cluster with a depth ranging from 6 km to 10 km (cross-sections of Figure 10). The Baranello sequence was formed by 340 events covering a time span from January to June 2016. Only five aftershocks show a magnitude $M_L$

larger than 3.0 (range between $M_L$ 3.0–3.6). Hypocentral depths range from 4 km to 13 km (Figure 11). Cross-section CD in Figure 11 shows a cluster delineating a plane dipping to the NE with an angle around 53°. The largest number of aftershocks were concentrated in the first 20 days of the sequence, but from 3 to 12 April 2016 there was a renewal of the seismicity with 51 events.

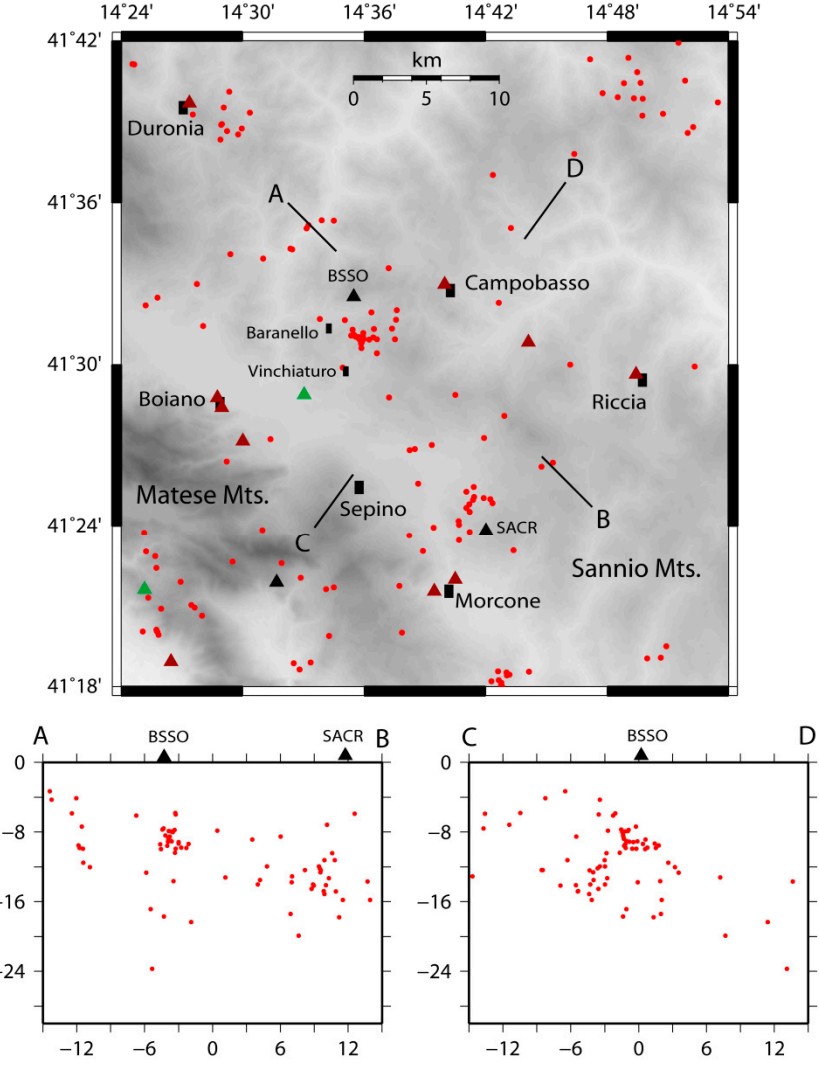

**Figure 10.** Map and cross-sections of the seismicity that precedes the 16 January 2016 main shock $M_w$ 4.3 of Baranello.

Slightly to the south, close to the locality of Vinchiaturo, a swarm-like sequence occurred (Figure 11). It started on 8 January 2017, with a magnitude $M_L$ 2.4 at 11 km of depth. An earthquake with $M_L$ 3.1 occurred on 12 January 2017 at the same hypocentral depth, which was the largest event in the swarm-like sequence. The swarm produced 300 earthquakes mostly concentrated in the time period from 8 January to 6 February 2017, with only three events reaching a magnitude of $M_L$ 3.0–3.1. This seismicity was reactivated from 9 to 17 June 2017, with maximum magnitude $M_L$ 2.7.

The first four days (8–11January) of the swarm are characterized by anomalously deep earthquakes (22–38 km of depth) with hypocenters not well constrained (57 earthquakes rejected; nearest station only at 23 km from epicenter; nearest station selection criteria). Unfortunately, four RSN permanent stations located within the first 20 km of the epicentral area were not working in those days. Figure 12 shows the comparison between hypocentral determination from the INGV bulletin (plot **a**) and those computed in this work (plot **b**). The histograms are related only to the seismicity of 29–30 December 2013 (Matese sequence; nearest station only at 16 km from epicenter) and 8–11 January 2017 (Vinchiaturo swarm;

nearest station only at 23 km from epicenter). The seismicity located in the 16–32 km depth range (INGV bulletin) belongs to the location not well constrained, due to the lack of stations in the epicentral area for both of these sequences.With these two histograms, we want to highlight the importance of seismic stations closest to the epicenterin the earthquake location. The lack of these close stations often leads to great indeterminacy in the hypocentral depth.

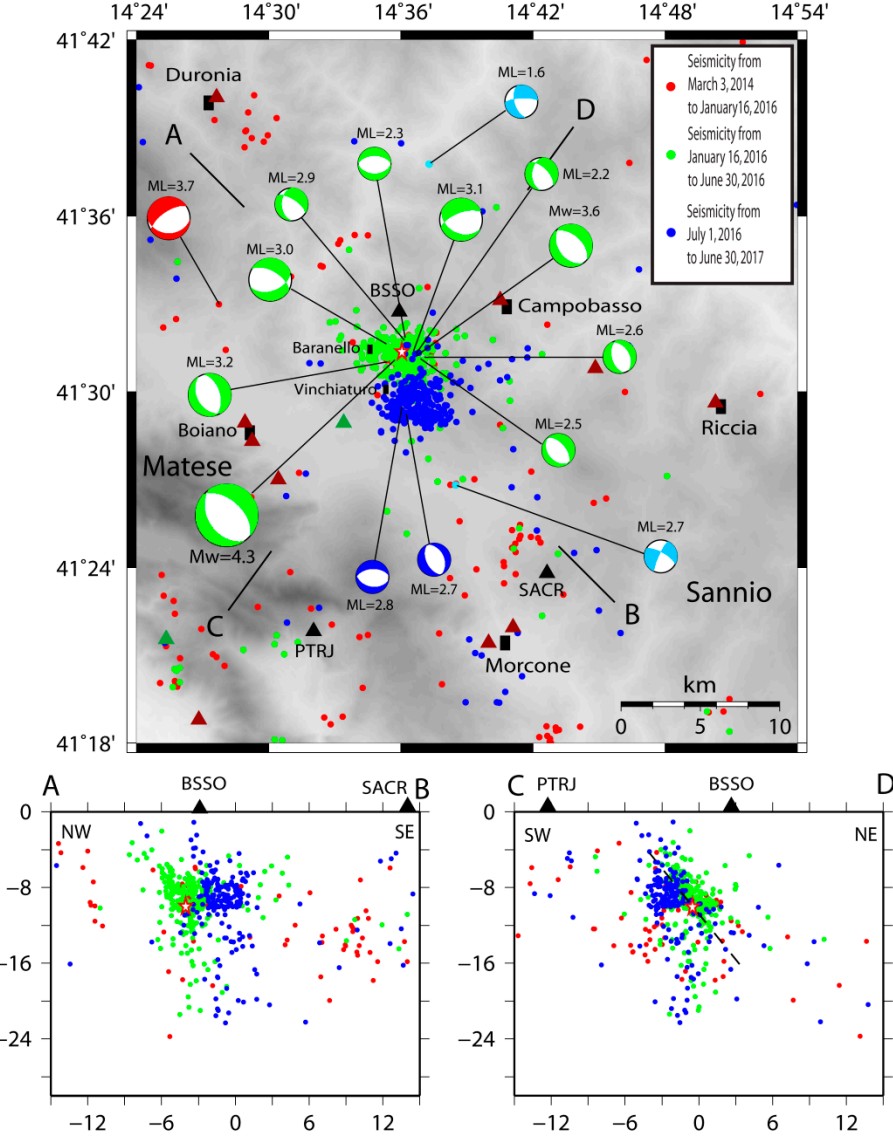

**Figure 11.** Map, cross-sections and fault plane solutions of the seismicity of the Baranello–Vinchiaturo area from 3 March 2014 to 30 June 2017. Star indicates the 16 January 2016 main shock M$_w$ 4.3.

The upper crustal seismicity shows a seismic structure dipping to the NE with an angle around 55°, very similar to the plane geometry in the Baranello sequence. Figure 11 gives an overview of the geometric relationships between the three seismic periods.

We computed 10 focal mechanisms for the Baranello sequence and 2 for the Vinchiaturo swarm, by using the first motion polarity of *P*-wave (Figure 11). The T-axis of these normal solutions strikes both along N–S and NE–SW directions. There are also focal mechanisms of two small events belonging to the July 2016 to September 2018 period, located close to the epicentral Baranello–Vinchiaturo area. These normal solutions both show a large strike–slip component.

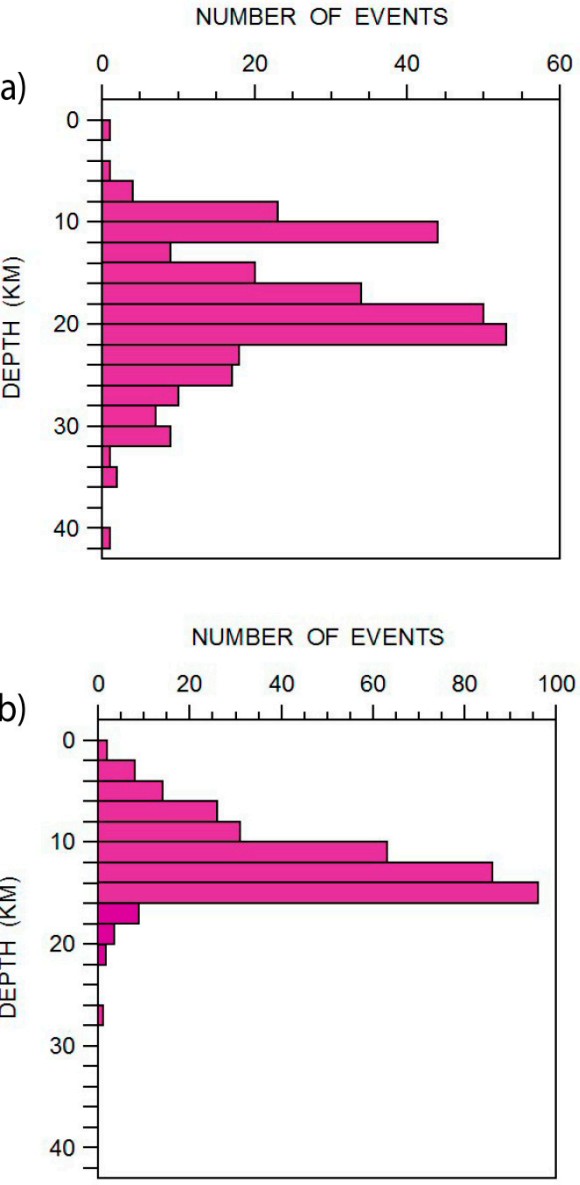

**Figure 12.** Hypocentral distribution of the Matese 2013–2014 sequence (only 29–30 December 2013) and of the Vinchiaturo swarm (only 8–11 January 2017): plot (**a**) from the Istituto Nazionale di Geofisica e Vulcanologia (INGV) bulletin and plot (**b**) in this work.

Figure 11, shows the fault plane solution of one deep event recorded in the study period (red beach-ball). It is an earthquake with $M_L$ 3.7 at 335 km depth detected in the Macchiagodena sector (CB). This rare deep earthquake beneath the central–southern Apennines is the seismological evidence of the activity of the Adriatic plate characterized by down-dip compression in the 165–370 km depth range [66].

### 3.3. The Beneventano and San Marco la Catola Seismicity

In this paper, we also analyzed the swarm activity that occurred during the study period in the Beneventano, an area bordering to the east and southeast the Matese massif (Figure 13). The main cluster is located near Pontelandolfo in the southeastern margin of the Matese Range. Seismicity started on 25 September 2014, with a magnitude $M_L$ 3.0 and lasted only a few days. In the NW–SE cross-section of Figure 13, we observe a sub-vertical cluster between 6 km and 14 km of depth. Fault plane solutions of the Pontelandolfo swarm show a dip–slip normal fault with T-axis NE–SW trending.

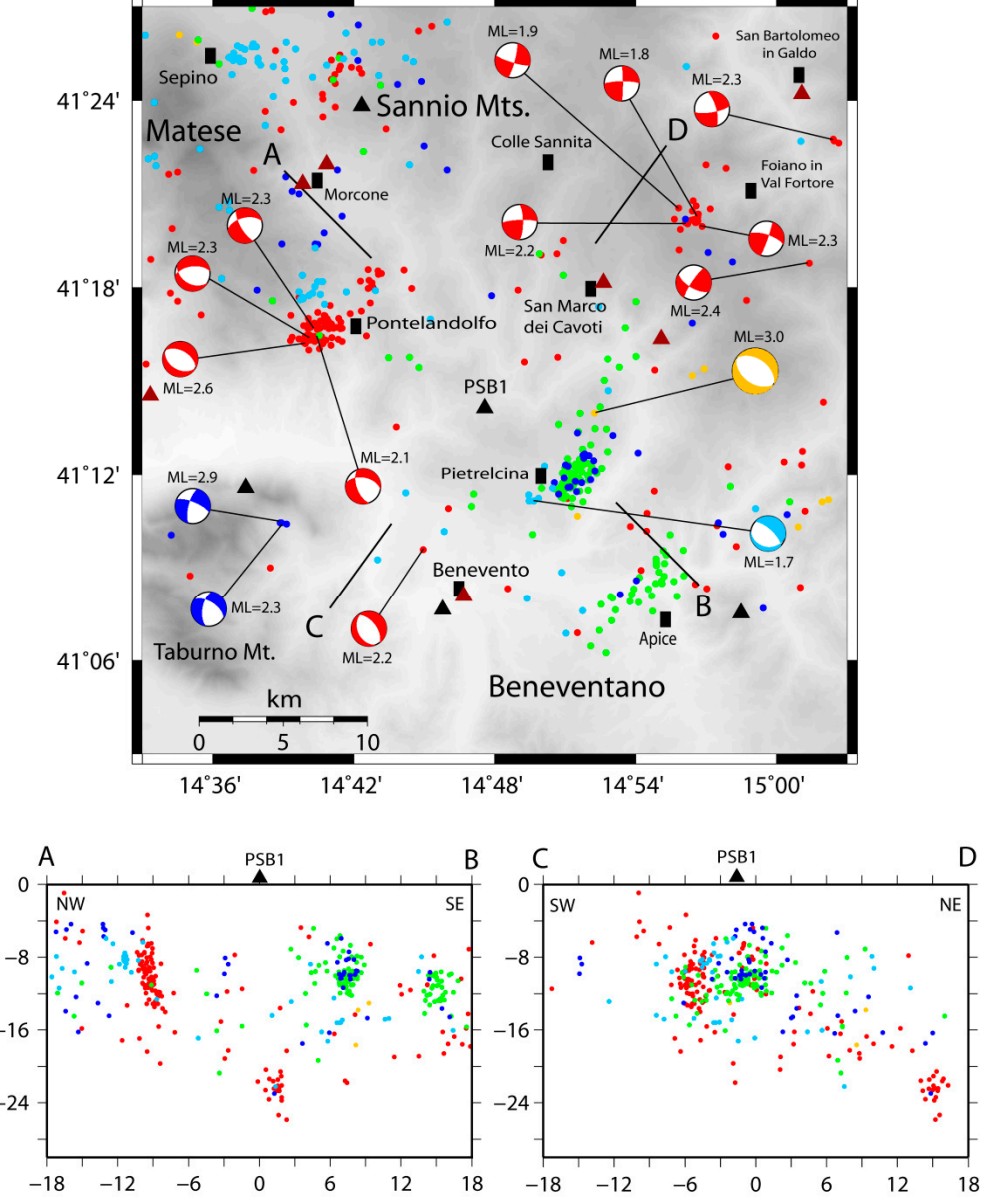

**Figure 13.** Map, cross-sections and fault plane solutions of the Beneventano area seismicity. For colored dots see legend of Figure 5.

In the same days, a small cluster was activated near Foiano in Val Fortore with maximum magnitude $M_L$ 2.3. This seismicity is deeper with ahypocentral depth ranging between 20 km and 28 km (see cross-sections in Figure 13) and withstrike–slip focal mechanisms.

In the period between 16 January and 30 June 2016, two swarms occurred in the southeastern part of the Beneventano area (green dots in Figure 13). Both swarms (Pietrelcina and Apice) show a sub-vertical trend with a depth ranging between 6 km and 12 km. These two swarms, with the Pontelandolfo and the Matese 2013–2014 sequences, are within the axial seismogenic belt of the southern Apennines. The Pietrelcina and Apice clusters are close to the April 1990 Benevento low energy sequence [67], characterized by hypocenters within the first 15 km of depth and by focal mechanisms with T-axes NE–SW oriented.

In the San Marco la Catola area, at the border between Puglia and Molise (Figure 14), in the period between March 2014 and September 2018, a dense cluster with 142 events occurred, with a maximum magnitude $M_L$ 3.5. This cluster, roughly WNW–ESE oriented, was characterized by deeper seismicity (depth range 14–28 km) and by strike–slip fault-plane solutions similar to the 2002 San Giuliano di

Puglia sequence located around 15 km to the north [42]. Probably, as with the 2002 sequence and the 1990–1991 Potenza sequence [43,68,69], the east–west nodal planes of these solutions are faults indicating a right lateral strike–slip motion accommodating the differential slab retreat, with the northern areas retreating eastward faster.

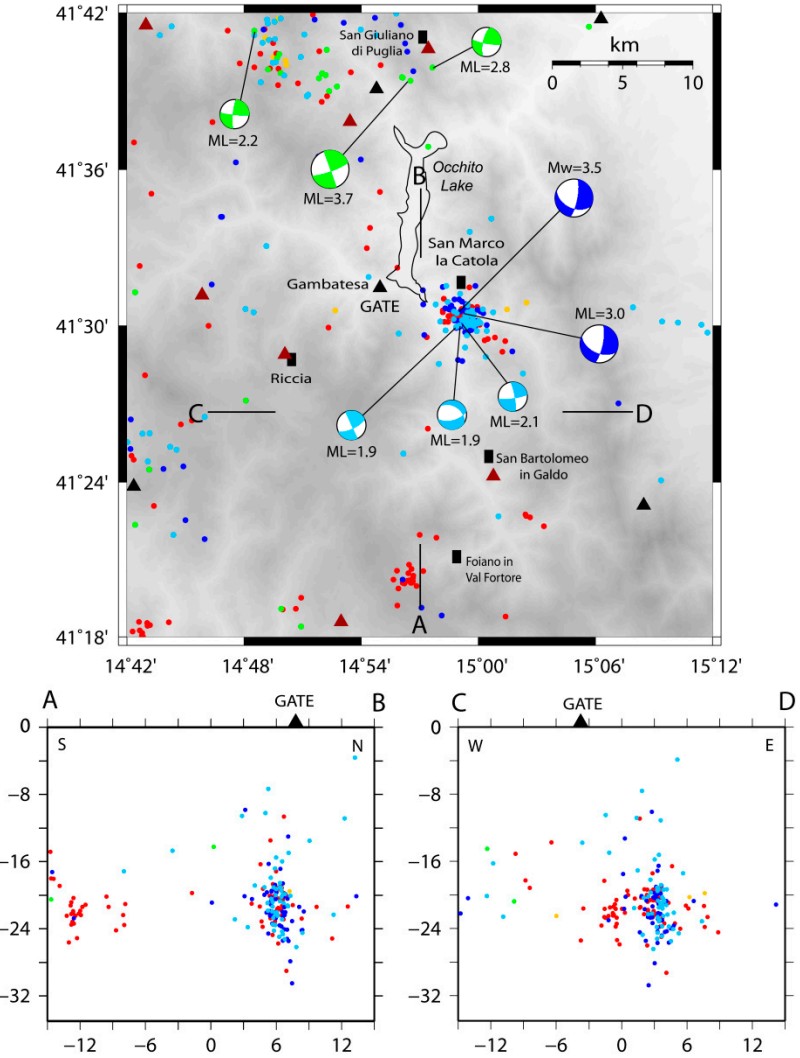

**Figure 14.** Map, cross-sections and fault plane solutions of the San Marco la Catola cluster. For colored dots see legend of Figure 5.

## 4. Discussion

### 4.1. The Matese Sequence Hypocentral Determination: Seismicity within the Brittle Upper Crust

A large amount of high-quality seismic data analyzed in this study and the 1D *MATESE* velocity model computed with the Velest code for this area of the central–southern Apennines, provide new constraints to the determination of hypocentral depths. We relocate 878 events of the Matese 2013–2014 sequence and do not find evidence of anomalously deep seismicity beneath the Matese massif as observed by Ferranti et al. (2015) [70] and Di Luccio et al. (2018) [71]. Ferranti et al. (2015) [70] observe in a cross-section with projected hypocenters of the Matese sequence, that the upward projection of the relocated seismicity intersects the earth surface very close to the eastern segment of the Bocca della Selva Fault (BSF). This fault may be an inherited structure from an older deformation stage. The hypothesis that it is occasionally activated during infrequent earthquakes should not be discarded. The authors used a dataset of 250 aftershocks and they show hypocenters concentrated in the depth

range of 10–20 km, with main shock hypocenter at 18 km. With their results they observe a new structure within the crystalline basement, above the mantle wedge, between 16 km and 20 km of depth, named the Matese fault.

Di Luccio et al. (2018) [71] observe hypocenters concentrated at a depth between 10 km and 25 km in the inferred crystalline basement of the chain for the Matese 2013–2014 sequence, with a main shock $M_w$ 5.0 located at 22 km of depth. They use a dataset of 350 aftershocks. Moreover, they identify a burst-like seismic sequence characterized by low-frequency content for the main shock and a dike-like aseismic volume enclosed by this seismicity. The estimated aseismic volume is approximately 30 km$^3$ and they interpret it as a crustal portion partially melted by an intrusive body. This burst-like rupture process is similar to that observed in the fluid injection-induced seismicity. The authors identified, through geophysical and geochemical signature, a magma ascending in the continental crust, revealing a mechanism of pluton emplacement. However, our observations demonstrate that the depth hypocenters are rather the same as those of the previous seismicity in this area [35,72]. Hypocentral depths of the main shock and of the largest aftershock computed in this study are 12.3 km and 12.7 km, respectively. D'Amico et al. (2014) [62] calculated the moment tensor solutions for 31 earthquakes with $M_L$ between 2.5 and 4.9 of the Matese 2013–2014 sequence by performing broadband waveform inversion, using the cut and paste (CAP) method [73]. The authors observe a hypocentral depth of 12 km for the main shock $M_w$ 5.0 using the misfit error as a function of depth. Regional Centroid Moment Tensor (RCMT) solutions were routinely computed by INGV using the Italian National Broad-Band Seismic Network (http://www.eas.slu.edu/eqc/eqc_mt/MECH.IT/; [63]). The best fit as a function of depth of these RCMT solutions gives hypocenters around 15 km of depth for both the two largest Matese shocks. These robust solutions confirm the quality of the hypocentral determination of our whole dataset. Ferranti et al., (2015) [70] and Di Luccio et al., (2018) [71] were probably misled by the difficulty in reading the *S*-phases and the lack of stations (nearest station only at 15–16 km from epicenter) in the epicenter zone in the first 24 h of the sequence. After the installation of the first temporary stations in the epicentral area, no anomalously deep event is located, as shown in this study.

Important features are evident in the first hours after 29 December 2013 (17:08 UTC) main shock ($M_w$ 5.0). We observe that the hypocenters depict two clusters significantly elongated in a vertical direction, with the main shock located at 12.3 km of depth (Figure 8a). The "dike-like aseismic volume" [71] is filled by the aftershocks in the following days with a migration of the seismicity to the southeast, exactly in the area of the largest aftershock ($M_w$ 4.2,20 January2014). A characteristic of this sequence is the large concentration of aftershocks in the crustal part around the $M_w$ 4.2 event of San Potito Sannitico. Both events show a NW–SE striking normal fault mechanism (Figure 6). They occurred on two very close SW-dipping rupture planes (Figure 9a,b, cross-sections CD). Convertito et al. (2016) [74] investigated the source directivity of the 29 December 2013 main shock using a multiapproach analysis. They identify a dominant down-dip rupture direction toward S–SW according with the observed macroseismic intensity distribution that shows higher values southwest of the epicenter [75]. Valenteet al., (2017) [76] identified several ground effects of the $M_w$ 5.0 event. In particular, a secondary nature co-seismic scarp around 4.5 km from the epicenter resulted from gravitational adjustment of the water rich, fractured rocks of the damage zone associated with the fault that bounds the Mt. Airola basin to the SW. The spatial distribution of these ground effects covers an area of at least 90 km$^2$ [77]. The relocation of the aftershocks relative to the main shock (30–31 December 2013) in our work, depicts an area elongated S–SE with events located obliquely down-dip with respect to the main shock (double-difference location, Figure 9a, cross-section AB). Down-dip rupture propagation is generally not so common [78]. The down-dip propagation of the fracture explains the estimated relatively short rupture length (L = 2.1 km) and low rupture velocity ($V_r$ = 1.9 km s$^{-1}$) [74]. The static stress transfer of the main shock fault ($M_w$ 5.0) caused the large number of aftershocks located to the S–SE and the second large rupture within this sequence ($M_w$ 4.2, largest aftershock of 20 January 2014). The aftershock distribution of the first and the second part of

the sequence (Figure 9a,b) depicts a geometry similar to the seismogenic fault of *Aquae Iulie* in the northern part of the Matese massif, with a SW-dipping normal fault [10].

## 4.2. The Baranello and Vinchiaturo Seismic Sequences: ANE-Dipping Structure

With the Baranello and Vinchiaturo seismic sequences, in the northeastern Matese, we observe a change in polarity of faulting, with a seismic structure dipping to the NE. These two sequences are located within the southeastern edge of the 1805 Boiano fault ($M_w$ 6.6; Figure 15). The 1805 seismogenic fault shows a NE-dipping plane without surface manifestations [6,7,39]. Cucci et al., (1996) [6] proposed a model for the deformation field produced by this blind almost pure dip–slip normal fault through the drainage pattern study in the area, suggesting that the whole area reflects repeated 1805-type earthquakes along the same NE-dipping fault.

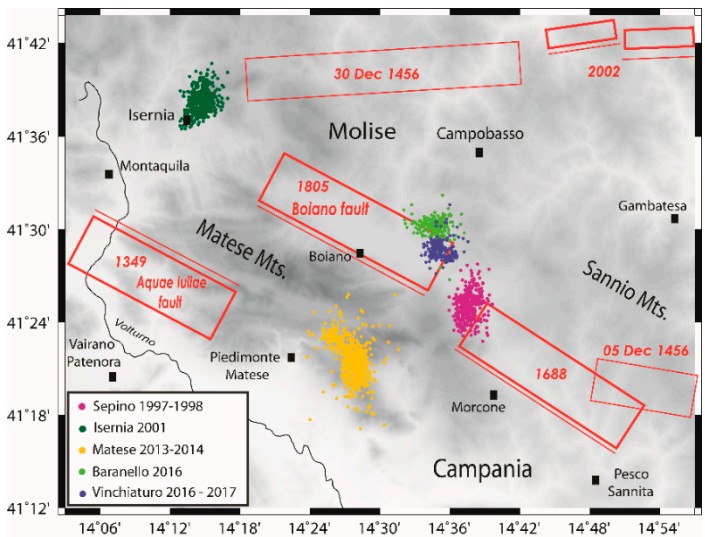

**Figure 15.** Historical seismogenic sources and most relevant instrumental seismicity from 1997 and 2017 in the Sannio–Matese area (modified from [70]). In the legend, colors are related to the seismicity.

Low magnitude seismic sequences and swarms occurred in the period 1997–2010 along the northern side of the Matese massif, with hypocenters within the uppermost 15 km of the crust. In particular, the 1997–1998 ($M_L < 4.2$) and 2001 ($M_L < 3.6$) seismic sequence clusters are located at the E–SE and W–NW lateral tips, respectively, of the 1805 seismogenic source (Figure 15; see [35,72,79,80] for details on these sequences). The 1997–1998 sequence [79,81] is located close to the southeast of the Baranello–Vinchiaturo sequences (2016–2017) and between the seismogenic sources of the 1805 and 1688 earthquakes (Figure 15). The 1997–1998 and 2001 seismic sequences highlight the existence of small N–S to NNE–SSW striking ruptures that belong to a secondary longitudinal extension affecting the chain (NNW–SSE extension, [72]). In this kinematic picture the eastward migration of the chain is responsible for the NW–SE striking large earthquake faults, whereas the curvature and thinning of the chain could cause small activity on the NE–SW striking faults [82].

The main shock of the Baranello sequence shows a hypocentral depth of 10 km. Our relocations are in agreement with the best fit as a function of depth of the Regional Centroid Moment Tensor (RCMT) solution routinely computed using the Italian National Broad-Band Seismic Network of the INGV (http://www.eas.slu.edu/eqc/eqc_mt/MECH.IT/; [63]). Focal mechanisms of the Baranello and Vinchiaturo clusters suggest the presence of both active local NE–SW and N–S extension (Figure 11). Detailed topographic profiles across the active fault segments of the Bojano basin provided post-LGM (15 ± 3 kyr) slip rates up to ~2 mm/yr [37]. The interseismic GPS velocity field across the transition zone between central and southern Apennine has recently been investigated [83]. The authors show that the extension rate computed across the Matese massif along an anti-Apennine profile is 2.0 ± 0.2 mm/yr.

The interseismic velocities projected along the profile show that the maximum extension does not follow the topographic high of the southern Apennines but is shifted toward the eastern outer belt. This result confirms that of historical seismicity which highlights the presence of NE-dipping seismogenic structures responsible for the largest earthquakes in the southern Apennines (1456,1688, 1702, 1732,1805; [5,7,38,84] located on the Adriatic side of the mountain range. These observations confirm the progressive eastward migration of the active belt that is due to nucleation of younger splay faults and to the deactivation of the break-away zone and the western faults of the system [24].

### 4.3. San Marco la Catola and Foiano in Val di Fortore Clusters: Evidence of Right-Lateral Strike-Slip Kinematics

The almost pure right lateral strike–slip motion earthquakes of the San Marco la Catola and Foiano in Val Fortore clusters (Figures 13 and 14), which occurred just east of the axial seismogenic belt of the southern Apennines, have a hypocentral depth ranging between 14 and 28 km.

Similar seismicity, with right lateral strike–slip kinematics and deep hypocenters, was observed with the Monteleone di Puglia ($M_L$ 4.9) earthquake of the 6 May 1971 [85] (Gasparini et al., 1985), during the two sequences that struck the region around Potenza (Basilicata) between 1990 ($M_L$ 5.7) and 1991 ($M_L$ 5.2; [68]) and with the 27 September 2012, $M_L$ 4.1 Benevento earthquake [86]. Moreover, pure right-lateral slip between 10 km and 24 km depth over a nearly vertical east–west fault occurred in the 2002 San Giuliano di Puglia (eastern Molise) sequence [42] located 18 km to the north of San Marco la Catola. A similar sequence with right-lateral strike–slip motion struck the Molise (Montecilfone-Larino) in August 2018 ([87]; Frepoli et al., in preparation). The San Marco la Catola and Foiano in Val Fortore clusters, along with all the seismicity listed above, are located within the Apulian crust, in the footwall of the easternmost Apennine thrust and just below the bottom of the Apulian carbonates, within the middle crust [69,86]. In fact, the Apenninic area where the Apennine units overthrust the Apulian crust is characterized by an articulated rheological layering, with two brittle layers located at different depths: a brittle upper crust (9 km to 12 km thick), with the Apennine extensional seismicity, overlaying a ductile middle and lower crust, and a thick brittle, possibly seismogenic crust (21–23 km thick, the Apulia foreland including the sedimentary cover and part of the middle crust) which overlies a ductile lower crust [69]. We have in this area the coexistence of two different stress regimes, vertically overlapping, a shallow extensional one (<15 km) and a deep transcurrent one (20–25 km) with a common SW–NE direction of the $\sigma_3$ principal stress [69,86]. The seismotectonic environment of these sequences is of regional importance. In fact, the extension of the area from Montecilfone-Larino to Potenza in a NNW–SSE direction is around 150 km, comprising the whole Apulian foreland crust. These crustal-scale E–W striking fault zones are structures that were possibly inherited from previous tectonic phases and reactivated under the present-day deformation field. These fault zones are strongly dependent on the rheological stratification [69]. East–west right-lateral strike–slip faulting is kinematically consistent with the widely documented differential slab retreat diffused in the Apulian region and seismicity is favored at middle-to-lower crustal depth by the existence of hydrated layers in the lower crust [88].

From a seismotectonic point of view, this kinematics might be shared by several much larger earthquakes that took place on the Adriatic side of the southern Apennines. In fact, many authors ([44], and references therein) considered these structures as potential seismogenic sources of moderate-to-large magnitude events. The crustal structure of these deep strike–slip seismogenic sources is scarcely known. In the current debate, E–W strike–slip faults have been interpreted either as second-order transfer faults between NW–SE trending master normal faults, or as primary lithospheric ruptures of the Adria plate [11,34,89–92]. The fault plane solutions computed in this study provide a useful tool for further studies and may yield additional constraints on regional geodynamic and seismotectonic models.

## 5. Conclusions

In this study, we have computed the largest earthquake catalog for the moderate-to-low magnitude normal faulting events of the Matese sequence 2013–2014 ($M_w$ 5.0). The high number of relocated events using an accurate 1D velocity model and an appropriate $V_p/V_s$ ratio provide new constraints on the hypocentral depth determination. The best fit as a function of depth of the RCMT solutions routinely computed by INGV by using the Italian National Broad-Band Seismic Network, gives a depth for the mainshock and the largest aftershock that is very similar to theestimate of this study. No anomaly in the hypocentral depth computation is observed. We exclude active volcanic and intrusive processes within the lower crust. Our results are in agreement with the brittle behavior of the upper crust as seen in previous seismological studies of the central–southern Apennines. In fact, most of the Apennines earthquake energyrelease concentrates at a depth less than 12 km within the axial zone of the chain. Moreover, there is not a dike-like aseismic volume enclosed by this seismicity. The 878 relocated aftershocks fill all the upper crustal sector involved in the sequence with a large concentration of events in the southernmost part, close to the largest aftershock ($M_w$ 4.2) of 20 January 2014.

We observe that the lack of seismic stations within a radius of 16–20 km from the epicentral zone could lead to hypocentral determinations that are not well constrained. This is very evident for the relocations of the first two days of the Matese sequence (29–30 December 2013), with the nearest station at 15–16 km from epicenter, and for the first part of the Vinchiaturo swarm (8–11 January 2017), with the nearest station at 23 km from the epicenter. In fact, the lack of stations near the epicenter and therefore lack of *S*-wave first arrival times, do not allow for a correct determination of thehypocentral depth.

In addition, we analyzed the seismicity occurred in the Matese and Beneventano areas since 2014 until September 2018. Relatively small E–W trending clusters are found within a crustal volume extending between 14 km and 28 km depth in the eastern peripheral zones of the Apenninic belt. This seismicity shows right-lateral strike–slip kinematics similarly to the Potenza (1990–1991), Molise (2002) and Benevento (2012) sequences. This is a clear difference in seismogenic behavior between the axial zone of the Apennines, characterized by NW-trending normal faulting, and the external part of the belt.

**Author Contributions:** Data curation, B.T. and A.F.; Investigation, B.T., A.F. and G.D.L.; Methodology, A.F., G.D.L. and P.D.G.; Software, P.D.G.; Supervision, C.D. All authors have read and agreed to the published version of the manuscript.

**Funding:** This research received no external funding.

**Acknowledgments:** We are grateful to the Assistant Editor Scofield Chen, to Giusy Lavecchia and to three anonymous reviewers for the useful and constructive comments that helped to improve the original manuscript. A special thanks to Alessandro Marchetti for the help given in the HypoDD analysis and the elaboration of some figures of this workand to Stephan Monna for improving the English language of the manuscript.

**Conflicts of Interest:** The authors declare no conflict of interest.

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
