# Peer review of "The 2013–2018 Matese and Beneventano Seismic Sequences (Central–Southern Apennines): New Constraints on the Hypocentral Depth Determination"

_geosciences, doi:10.3390/geosciences10010017_

Round 1

Reviewer 1 Report

The Authors present high-quality relocation of low-to-moderate seismic sequences occurred from December 2013 to September 2018 in the Matese and Beneventano areas, in the transition zone between the central and southern Apennines. The elaborated dataset consists of 2378 ipocentral locations and of 55 new first-motion focal mechanisms. Most of the events are extensional, and a few of them have strike-slip kinematics. All of them are substantially coherent with a SW-NE extensional axis.

The main value of the paper is just in the presentation of the new high quality hypocentral dataset and the new focal mechanisms. As a whole, the paper is not really stimulating as it does not go in deep with the seismotectonic interpretation of the sequences and with the possible association with active individual structures. But, in effect, this was not the target of the paper, which in my opinion can be published after a substantially moderate revision.

The “Introduction”,   the “Geologic and Seismotectonic setting” , as well as the Discussion, do not take in consideration all the available literature. In particular, the authors miss to cite and discuss some relevant papers on the strike-slip deformation which characterises the middle crust in the southern Italy (1990-91 Potenza sequence in e.g. Boncio et al., 2007) and specifically in the Beneventano area (Adinolfi et al., 2015). As well, they do not consider, two relevant papers on the geometry of the Quaternary and potentially seismogenic east-dipping extensional structures in the Sannio Matese and Beneventano area (Brozzetti, 2011; Ferrarini et al., 2017) .

I do not understand why the Authors associate the San Marco la Catola cluster to E-W strike -slip structure. In fact, at the light of the cross-sections in Figure 16, the high-angle hypocentral volume dips eastward. Therefore, the preferential seismic plane of the focal mechanism solutions in Figure 15 are the nearly N-S tending right-lateral ones.

The figure are somehow confusing and do not help the reader to follow the paper. Here a list of problems, with the figure

A-It is not easy to immediately understand the reciprocal geographic location of the studied seismic sequences and swarms. To solve such a problem, I suggest to add at the beginning of the paper a location map with the boundaries of all the rectangular study areas. Furthermore, at the end of the paper it would useful to have a map with all the studied sequences and swarms and with their representative focal mechanism, and depth range. This final figure should be accomplished by synoptic table with all information on the various sequences and swarms

B-It is important that composite figures (as an example Fig. 6 a,b,c ) are contained within one page and not in more than one pages.

C-Figure 7, with the time space evolution of the Matese sequence, is not readable: maps and sections are too small, too many and in too many pages. I suggest to zoom more on the sequence area and to reduce the number of maps , putting in a same map and corresponding sections more than one temporal interval, using colour and symbol to show time space evolution.

D-Fig. 8 is at the end of the paper , after Fig. 18, and not in between Fig. 7 and 9. Why? Remove the figure and the corresponding text form the end of the paper and discuss it after Fig. 7.

E- In Figure 18, the 2001 Isernia and 1997-98 sequences are reported. This is interesting , but additional information on them must be given in the text.

In conclusion, in my opinion, once the authors have taken into consideration the above comments and suggestions, the paper may be published in Geosciences. Hypocentral data set and focal mechanism parameters would be as well published as supplementary materials.

PAPERS to be cited

Boncio P., Mancini T., Lavecchia G., Selvaggi G. (2007). Seismotectonics of strike-slip earthquakes within the deep crust of southern Italy: Geometry, kinematics, stress field and crustal rheology of the Potenza 1990-1991 seismic sequences (Mmax 5.7). TECTONOPHYSICS, 445, 281-300, doi:10.1016/j.tecto.2007.08.016.

Adinolfi G.M., De Matteis R., Orefice A, Festa G, Zollo A, de Nardis R. & Lavecchia G. (2015)- The September 27, 2012, ML 4.1, Benevento earthquake: A case of strike-slip faulting in Southern Apennines (Italy). Tectonophysics, 660, pp. 35-46.

Ferrarini Federica, Paolo Boncio, Rita de Nardis, Gerardo Pappone, Massimo Cesarano, Pietro P.C. Aucelli, Giusy Lavecchia (2017)- Segmentation pattern and structural complexities in seismogenic extensional settings: The North Matese fault system (central Italy), Journal of Structural Geology, http://dx.doi.org/10.1016/j.jsg.2016.11.006.

Boncio P., Mancini T., Lavecchia G., Selvaggi G. (2007). Seismotectonics of strike-slip earthquakes within the deep crust of southern Italy: Geometry, kinematics, stress field and crustal rheology of the Potenza 1990-1991 seismic sequences (Mmax 5.7). TECTONOPHYSICS, 445, 281-300, doi:10.1016/j.tecto.2007.08.016.

BROZZETTI (2011) - The Campania‐Lucania Extensional Fault System, southern Italy: A suggestion for a uniform model of active extension in the Italian Apennines. TECTONICS, VOL. 30, TC5009, doi:10.1029/2010TC002794.

Reviewer 2 Report

Comment on “The 2013-2018 Matese and Beneventano seismic 2 sequences (Central-Southern Apennines): new 3 constraints on the hypocentral depth determination” Manuscript ID: geosciences-659873

Major comments

I cannot find Figure 8 in the manuscript. The author should use “main shock”. Do not use “mainshock”. “main” and “shock” should be separated. A professional company should check English of your manuscript. Please make some subsections in the section of “Discussion”. What you want to say is not clear in Discussion. So, divide “Discussion” into smaller sections and give them short titles. The short titles should show clearly what you want to say. For example,

Discussions

Distribution of accurate hypocenters Stress state estimated from focal mechanisms A new tectonic model in this region

And so on.

Minor comments

Line 131 “On December 30, 2013, one day after the Mw 5.0 San Gregorio Matese mainshock, were installed 131 two temporary stations in the epicentral area of the sequence …..”  This sentence has no subject.

Line 133 “In the same way, after the Mw 4.3 mainshock of Baranello, on January 17, 2016 was 133 installed one temporary station …”  This sentence has no subject.

Line 159 “azimuthal gap less than 180” Add a unit of 180 !. I guess 180°.

Line 159 “Due to the prevalence of upper crust seismicity (0-15 km) in our dataset” What do you mean by this sentence? Please describe more clearly.

Lines 249-252 “we observe many low magnitude aftershocks located in the 16-22 km depth range with small hypocentral errors (quality A of Hypoellipse) but with the nearest station (VAGA) only at 15-16 km from the epicenter, and for this reason rejected.” What do you mean by this sentence? What did you reject? Please describe more clearly.

Line 273 “The second group consist in 196 events” --> “The second group consists of 196 events”

Table 2 Add a unit of Qp, I guess delta s, d, and r are in degree. 20°, 20°-40°, and so on.

Line 298 “This mainshock was preceded”  not “mainshock”, but “main shock”

Line 307 “occurred a swarm-like sequence” à “a swarm-like sequence occurred”

Lines 308-311 “The January 12, 2017, seismicity reached the maximum magnitude ML 3.1 with an earthquake at the same hypocentral depth.” à “An earthquake with ML = 3.1 occurred on January 12, 2017 at the same hypocentral depth, which was the largest event in the swarm-like sequence.”

Lines 311-312 “There was a renewal of this seismicity from 9 to 17 June 2017” à “This seismicity was reactivated from June 9 to 17, 2017”

Lines 330-332 Figure caption of Figure11 “With these two histograms we want to highlight the importance of the epicenter closest seismic stations in the earthquake location. Their absence often leads to greater indeterminacy in the hypocentral depth.” These sentences should not be described in the figure caption, but in the text.

Lines 336-337 “We computed 10 first motion fault plane solutions for the Baranello sequence and 2 for the 336 Vinchiaturo swarm” à “We computed 10 focal mechanism solutions for the Baranello sequence and 2 for the Vinchiaturo swarm by using the first motion polarity of P-wave.”

Line 347 “the only deep event” à “one deep event”

Figure 15 The focal mechanism solutions in Figure 15 should be plotted on Figure 14. If you do so, you don’t need Figure 15.

Figure 17 The focal mechanism solutions in Figure 17 should be plotted on Figure 16.

Reviewer 3 Report

Interesting study, which provides new information on active structures in Italy. High quality data are included in its hypocentral determination, taken from the permanent seismic network and temporary stations and the seismograms were chopped again. The conclusions of the work are interesting and reveal that the context of seismicity is fragile and is not related to other types of internal processes.
The methodology is correct. The wording is correct and reads well, but the presentation of results can be summed up a bit.
The figures must be substantially improved and their number reduced. In this regard, some suggestions are made.

Suggestions:

Indicate in the methodological section what software has been used in the mincing of the phases and how the error has been estimated (I understand that it has been the Hypoellipse) Figure 1. Black dots do not look good and are confused with the background. You can put a lighter background. The inset is too small in size. The font size and type should be homogenized. It is advisable to merge figures 1 and 2. In figure 2 the main faults could be represented. The gray background does not provide any information. The inset must be deleted. The first part of Table 1 should be deleted and cite those authors in the text. This information is also redundant with Figure 4a. Figure 4b could be dispensable. Figure 5 could show the main faults in the area. The gray background does not provide any information. This figure could represent only the study area.The meaning of the triangles must be included in the legend. In Figure 6 the map can be made larger. The caption should indicate whether or not all events are projected in the cuts. Parts b and c of the figure can be placed next to each other to take advantage of the space. The gray background does not provide any information. In figure 7 you could put all the information on the same map, larger. The profiles (larger) could be kept at the bottom. It is too large a figure and it is suggested to remake it. Figure 9 could be merged with Figure 6, since they are the same except the focal mechanisms. Figure 8 is badly placed in the manuscript, it must go before 9 and not after 17. Like the 7 put all the information on the same map. In figure 10 you could put all the information on the same map, larger. The profiles (larger) could be kept at the bottom. It is too large a figure and it is suggested to remake it. Figure 11. In the caption you should put what you see in the figure, not a text with the interpretation of it. Figures 12 and 13 can be merged into one. The map can go larger. The gray background does not provide any information. Figure 14. Indicate what the colors of the dots correspond to. It can be merged with 15. Figure 16. Indicate what the colors of the dots correspond to. It can be merged with 17. Figure 18. Indicate in the map legend what each color corresponds to. Minor error: line 165. "(Kissling et al., 1994) starting", a space is missing.
